# Changes in the Fungal Community Assembly of Apple Fruit Following Postharvest Application of the Yeast Biocontrol Agent *Metschnikowia fructicola*

**Antonio Biasi** [1] , **V. Yeka Zhimo** [1] , **Ajay Kumar** [1] , **Ahmed Abdelfattah** [2] , **Shoshana Salim** [1] , **Oleg Feygenberg** [1] , **Michael Wisniewski** [3] **and Samir Droby** [1,*]

1 Department of Postharvest Science, Agricultural Research Organization, The Volcani Center, P.O. Box 15159, Rishon Lezion 7505101, Israel; antoniobiasi84@gmail.com (A.B.); yekaz@volcani.agri.gov.il (V.Y.Z.); ajayk@volcani.agri.gov.il (A.K.); shoshi@volcani.agri.gov.il (S.S.); fgboleg@volcani.agri.gov.il (O.F.)

2 Institute of Environmental Biotechnology, Graz University of Technology, Petersgasse 12, 8010 Graz, Austria; ahmed.abdelfattah@tugraz.at

3 Department of Biology, Virginia Polytechnic Institute and State University, Blacksburg, VA 24060, USA; wisniewski@vt.edu

* Correspondence: samird@volcani.agri.gov.il

**Abstract:** Recently, increasing focus has been placed on exploring fruit microbiomes and their association with their hosts. Investigation of the fruit surface microbiome of apple has revealed variations in the composition and structure depending on management practices, phenological stages, and spatial distribution on the fruit itself. However, the fate of the fruit surface microbiome assembly and dynamics in apple following interventions such as the application of biocontrol agents remains unknown. The objective of the study was to explore the effect of a postharvest application of a yeast biocontrol agent, *Metschnikowia fructicola,* on the composition of the epiphytic fungal microbiota on apples during cold storage. Our results demonstrated that the applied biocontrol agent, *M. fructicola*, persisted in high abundance (>28% relative abundance) on the fruit surface throughout the storage period. The biocontrol application significantly decreased the richness and caused a significant shift in the overall composition and structure of the fungal microbiome relative to untreated or water-treated controls. The yeast application reduced the abundance of several apple fungal pathogens, namely, *Alternaria*, *Aspergillus*, *Comoclatris*, *Stemphylium*, *Nigrospora*, *Penicillium*, and *Podosphaera*, throughout the cold storage period.

**Keywords:** disease management; fungi; microbiome; apple; biocontrol; postharvest

## 1. Introduction

Microorganisms represent an important component of fruit and their presence as epiphytic and endophytic communities has been documented in the literature [1–5]. However, their function in determining fruit quality and its susceptibility or tolerance to the development of postharvest fruit decay pathogens remains largely unexplored [2,3]. In this regard, high-throughput sequencing technologies are now allowing us to investigate the complexity of the composition and dynamics of microbial communities in different plant tissues and better understand their functions and interactions within the communities and with the host plant.

Interactions of fruit with its resident microbiota before and after harvest may have an important impact on fruit quality and shelf life. The host–microbiome interactions could be deleterious and result in the infection and establishment of postharvest pathogens, resulting in subsequent crop losses, which is a primary concern of fruit producers and exporters. Proper management practices for the control of postharvest pathogens are considered to be a cornerstone of profitable fresh fruit and vegetable production and supply chain systems [1–4]. Suggested options for postharvest treatments include physical-, chemical-, and

biological-based treatments [6]. Strict regulatory requirements have been imposed on the continued use of chemical fungicides in fresh fruit crops [7]. A viable alternative based on the use of microbial biocontrol products is now available for the management of important postharvest pathogens [8]. Among these products, Noli (Koppert Inc., Netherlands), based on the yeast *Metschnikowia fructicola*, is now registered in several countries and proved to be an effective antagonist of different pre- and postharvest pathogens [5,9–11]. Enhanced efficacy of *M. fructicola* was achieved when the yeast biocontrol agent was applied in conjunction with either hot water or steam treatments [11,12]. The mechanism of action by which *M. fructicola* exerts its biocontrol effect involves multiple effects that result in inhibition of fungal pathogens [11–14]. These include competition for nutrients and space; stimulation of the immune system in the host through overexpression of genes involved in defense signaling; and production of superoxide anions, chitinase, and pulcherrimin for iron sequestration [15–18]. The interaction of *M. fructicola* with the fruit microbiome, however, remains largely unknown. In a recent study, Zhimo et al. [5] demonstrated that a near harvest application of *M. fructicola* on strawberry fruit significantly altered the structure and composition of epiphytic microbial communities in the fruit. The application of *M. fructicola* caused an increase in bacterial and fungal taxa that are known to be beneficial microorganisms, suggesting that this could be an additional mechanism by which *M. fructicola* inhibits postharvest pathogens.

Numerous studies that have investigated the structure and composition of the microbiome on several fruit species have focused on shifts in the microbiome due to management practices, phenological stages, and their spatial distribution on the fruit itself [2,3]. Investigations of the assembly and dynamics of the fruit surface microbiome in apple following interventions such as application of biocontrol agents have not been previously reported. Such studies are important for providing information that is necessary for rational decision making about the timing of application of the biocontrol agents and possible understanding of the overall microbial community dynamics that may lead to suppression or proliferation of fungal pathogens after harvest.

The present study addressed the following objectives: (1) to determine the effect of a postharvest application of *M. fructicola* on the composition of the epiphytic apple fruit fungal microbiota during cold storage and (2) to study the population dynamics of specific known apple fruit pathogens during storage following the application of *M. fructicola* after harvest. Our results demonstrated significant differences in alpha and beta diversity between fruit treated with the biocontrol agent relative to untreated or water-treated controls. *M. fructicola* persisted on the fruit surface and caused major shifts in the composition of postharvest pathogens while reducing the abundance of apple fungal pathogens throughout the cold storage period.

## 2. Materials and Methods

### 2.1. Fruit Treatment with the Biocontrol Agent

Apple fruits of the cv. Pink Lady (*Malus domestica*) were harvested from a commercial orchard located in the northern Galilee region, Israel, (3380400400 N, 3582700400 E). Harvested fruit were transported to the laboratory on the same day and treated with *M. fructicola* (strain NRRL Y-27328, CBS 8853) [13]. Stock cultures of *M. fructicola* were stored at $-80\,^\circ$C in 20% of glycerol solution. To prepare the starter culture, 100 µL of the stock solution was transferred to a 100 mL Erlenmeyer flask containing 25 mL of yeast extract peptone dextrose broth (YEPD) and incubated on a shaker (200 rpm) at $28\,^\circ$C for 48 h. Then, 10 mL of the starter culture were transferred to a 1000 mL flask containing 200 mL of YEPD broth and incubated under the same conditions for 48 h. Yeast cells were subsequently harvested by centrifugation at $3000\times g$ for 10 min. The cell pellet was washed twice with sterile distilled water (SDW) and re-suspended in SDW to a final concentration of $10^8$ cells/mL. Fruit treatment with *M. fructicola* suspension ($10^8$ cells/mL) was done by dipping fruit for 2 min, followed by air drying, and finally moving the fruit to cold storage ($1\,^\circ$C). Two sets of controls were included, comprising of untreated fruits and those dipped



in sterile distilled water for 2 min. Each treatment group comprised three replicates with 100 fruits in each replicate (900 fruits in total).

## 2.2. Sampling, DNA Isolation, and Amplicon Sequencing

Fungal epiphytic microbiome sampling from the fruit was carried out four times—at 0 time (immediately after treatment and air drying), and at 2, 4, and 8 weeks of cold storage. At each sampling time and for each of the three treatment groups, sampling was done by swabbing the entire surface of each individual fruit (25 fruits per replicate) with sterile cotton swabs wetted with phosphate buffered saline (PBS) solution. The 25 swabs from each of the 25 fruits were combined to represent a replicate and placed in sterile falcon tubes containing 10 mL of sterile PBS buffer. Tubes were shaken at 250 rpm on a shaker for 20 min, followed by sonication in a water bath for 10 min to dislodge the microorganisms from the swabs. After aseptically removing the swabs, the supernatants were subjected to centrifugation at $3000 \times g$ for 10 min at 4 °C to obtain microbial pellets. Microbial DNA was then extracted from these pellets using DNeasy Power Soil Kit (Qiagen, Germany) following the manufacturer's instructions. Extracted DNA was then used for amplicon sequencing of ITS regions using the primers ITS3/KY02 and ITS4 [19] modified with Illumina adapters. PCR reactions were effectuated in a final reaction volume of 25 μL containing 12.5 μL 2X KAPA Hifi HotStart ReadyMix, (Kapa Biosystems, Wilmington, MA, USA), a final concentration of 0.4 μM of each primer, 2.5 μL of DNA template, and the remnant part with nuclease-free water. Amplification was performed in a T100 Thermal Cycler (Bio-Rad), with the following protocol: initial denaturation of 5 min at 95 °C, followed by 30 thermal cycles of 30 s at 95 °C, 30 s at 55 °C, and 30 s at 72 °C, and a final extension of 5 min at 72 °C. PCR products were cleaned using magnetic AMPure XP beads (Beckman-Coulter, Brea, CA, USA), and 50 ng of both amplicons of the same sample was pooled and Illumina sequencing libraries were constructed following the indications of the Illumina 16S Metagenomic Sequencing Library Preparations combined with the use of a Nextera Index Kit (Illumina, San Diego, CA, USA) containing 96 indexes. The quality of the library was then monitored by Agilent 2100 Bioanalyzer (Agilent), and the final paired-end sequencing of amplicons was performed on an Illumina MiSeq (Illumina) sequencer using a V3 600-cycle chemistry (Illumina, San Diego, CA, USA).

## 2.3. Data Analysis

QIIME 2 [20] was used to analyze data obtained from the Illumina Miseq platform. Fastq files were initially processed using the DADA2 algorithm [21], by implementing the default workflow (filtering, chimera identification and filtration, and merging of paired-end reads) with default parameters to obtain amplicon sequence variants (ASVs). The taxonomy assignment was performed using the naïve-Bayes classifier, included in DADA2, trained on the UNITE v7.2 [22] database for ITS with a 99% similarity threshold, and the final ASV table was obtained after removing singletons and ASVs assigned to other kingdoms different from Fungi. After normalizing the data to correct differences in sampling depth, all the downstream analyses were performed in R environment (R Version 3.6.2 in R Studio Version 1.2.5033).

The relative abundances of observed fungal taxa at the genera level were compared between the three treatment groups using bar plot charts made with the package phyloseq [23] in R. Alpha diversity (observed ASVs) of the fungal microbiome was compared between the treatments using a non-parametric Kruskal–Wallis test, with $p < 0.05$ considered as statistically significant. Pairwise comparisons between the three treatments were made using the Wilcoxon test, with $p < 0.05$ considered as statistically significant. Principal coordinate analysis (PCoA) based on Bray–Curtis dissimilarity along with permutational multivariate analysis of variance (PERMANOVA) using Adonis was performed to identify differences in fungal community composition between the treatments.

## 3. Results

### 3.1. Amplicon Sequencing Results

After quality filtering, as well as removal of mitochondrial and chloroplast sequences, singletons, and chimeras, the ITS data were normalized to 5174 sequences per sample to obtain 882 ASVs.

### 3.2. M. fructicola Treatment Altered the Diversity and Composition of the Apple Fungal Microbiome during Storage

Within-sample diversity richness (α-diversity) of the fungal microbiome of each sample was measured by calculating the observed ASVs and conducting a non-parametric Kruskal–Wallis test to analyze for differences between the treatment groups (*M. fructicola*-treated, untreated, and water-treated controls) (Figure 1A). An overall significant effect of treatment ($p = 8.3 \times 10^{-6}$) on α-diversity was detected. Pairwise comparisons between the three treatments revealed that *M. fructicola*-treated fruit samples had a significantly reduced richness relative to both untreated (Wilcoxon test, $p = 3.6 \times 10^{-5}$) and water-treated (Wilcoxon test, $p = 3.6 \times 10^{-5}$) control fruits. No significant difference in richness was observed between untreated and water controls (Wilcoxon test, $p = 0.93$). When richness was separately examined in each treatment group across the four storage periods, no effect on storage period was observed within any of the three treatment groups (Kruskal–Wallis test, *M. fructicola*-treated: $p = 0.19$; untreated control: $p = 0.056$; water-treated control: $p = 0.6$) (Figure 1B).

**A**

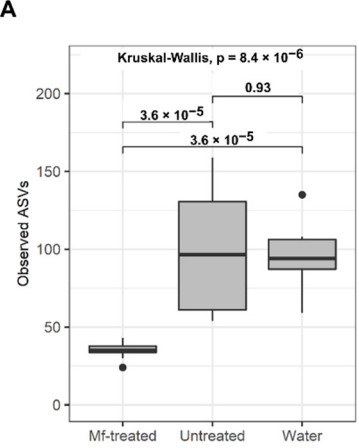

**B**

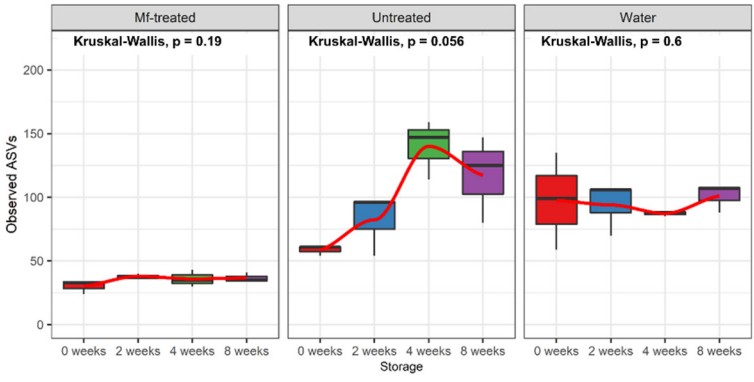

**Figure 1.** Boxplots of epiphytic apple fruit fungal richness (Observed ASVs) among *Metschnikowia fructicola*-treated and untreated samples. (**A**) Comparisons of the overall richness among *M. fructicola*-treated and untreated controls and (**B**) comparisons at different storage times for each treatment group.

A principal co-ordinate analysis (PCoA) paired with permutational multivariate analysis of variance (PERMANOVA) based on Bray–Curtis dissimilarity was used to test for differences in the overall community composition across the three treatment groups (β-diversity). *M. fructicola*-treated samples clustered separately towards the right of PCoA axis 1, while samples from the two controls clustered toward the left axis, indicating similarity in the microbial composition of the two control groups and a dissimilarity of the two groups from the *M. fructicola*-treated samples (Figure 2A). PERMANOVA results confirmed this finding by revealing a highly significant effect of treatment (adonis, $R^2 = 0.464$, $p = 0.001$) on the observed community structure variation among sample groups. β-diversity analyses were also conducted to assess the effect of storage time on the fungal microbiome composition separately in each treatment group, while PERMANOVA indicated that the variation of the community structure in all three treatment groups was significantly affected, but at different levels of significance in Figure 2B,C. The untreated control samples exhibited the greatest effect size ($R^2 = 0.746$, $p = 0.001$), followed by water-treated samples ($R^2 = 0.593$, $p = 0.001$), while the *M. fructicola*-treated samples were the least affected ($R^2 = 0.413$, $p = 0.003$) (Figure 2C).

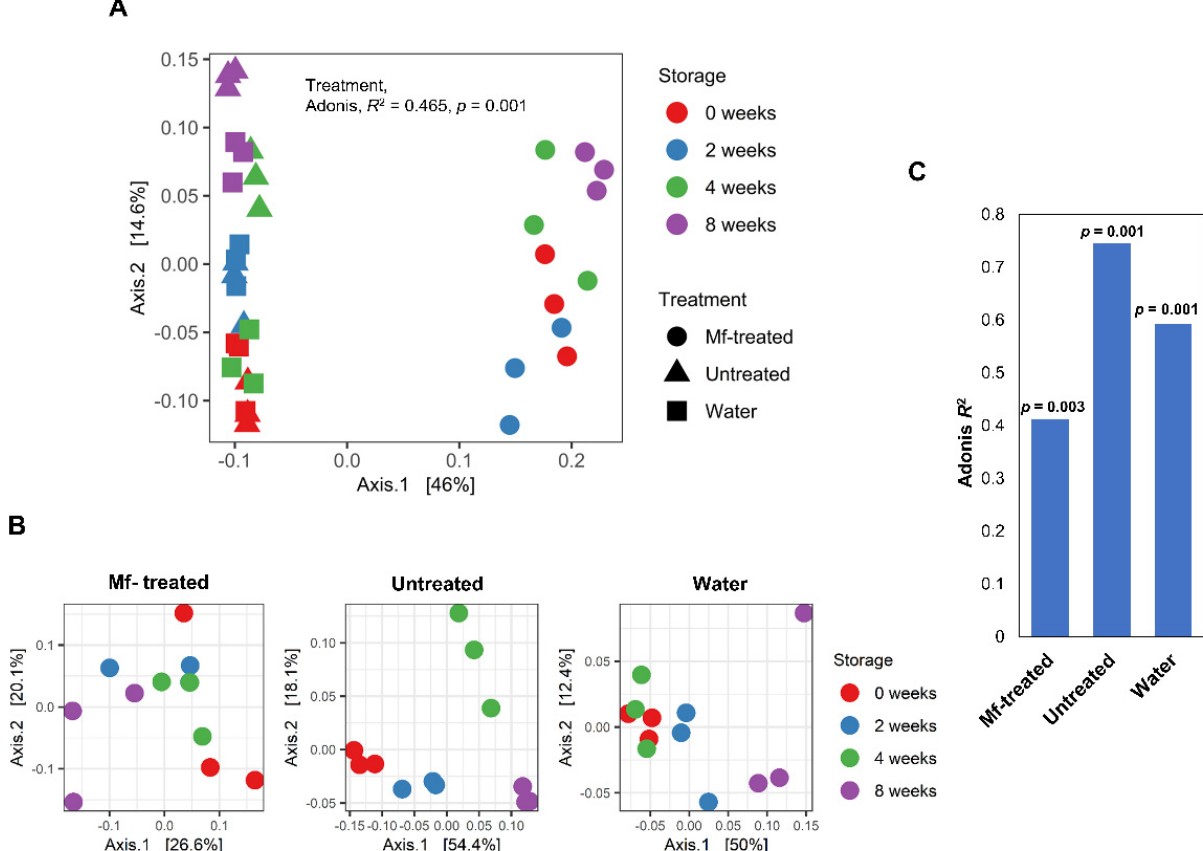

**Figure 2.** Principal coordinate analysis (PCoA) based on Bray–Curtis dissimilarity along with permutational multivariate analysis of variance (PERMANOVA) showing differences in fungal community composition between *Metschnikowia fructicola*-treated and untreated fruits. (**A**) PCoA plot visualizing clustering of treated samples away from control samples. (**B**) PCoA plots showing clustering of samples from different storage periods for each treatment groups. (**C**) Adonis $R^2$ values indicating the proportion of variances observed for each treatment group.

The relative abundance of the predominant fungal genera that were found to make up the apple carposphere fungal community in this study were obtained by considering those taxa that represent at least 0.1% of the total sequences. Bar plots visualizing the results showed that the distribution of these predominant genera differed between the three treatments, *M. fructicola*-treated, untreated, and water-treated, across all sampling

times (Figure 3). *Metschnikowia* was detected in high abundance and persisted throughout the examined storage period in *M. fructicola*-treated fruit (33.15% at 0 weeks, 28.05% at 2 weeks, 32.13% at 4 weeks, and 36.26% at 8 weeks). Other predominant genera included unidentified *Capnodiales* (47.03% in untreated, 35.12% in *M. fructicola*-treated, and 44.23% in water-treated samples), *Cladosporium* (30.17% in untreated, 23.60% in *M. fructicola*-treated, and 31.82% in water-treated samples), *Aureobasidium* (6.02% in untreated, 3.36% in *M. fructicola*-treated, and 6.83% in water-treated samples), unidentified *Pleosporaceae* (5.34% in untreated, 2.02% in *M. fructicola*-treated, and 5.0% in water-treated samples), and *Vishniacozyma* (1.84% in untreated, 0.95% in *M. fructicola*-treated, and 2.36% in water-treated samples).

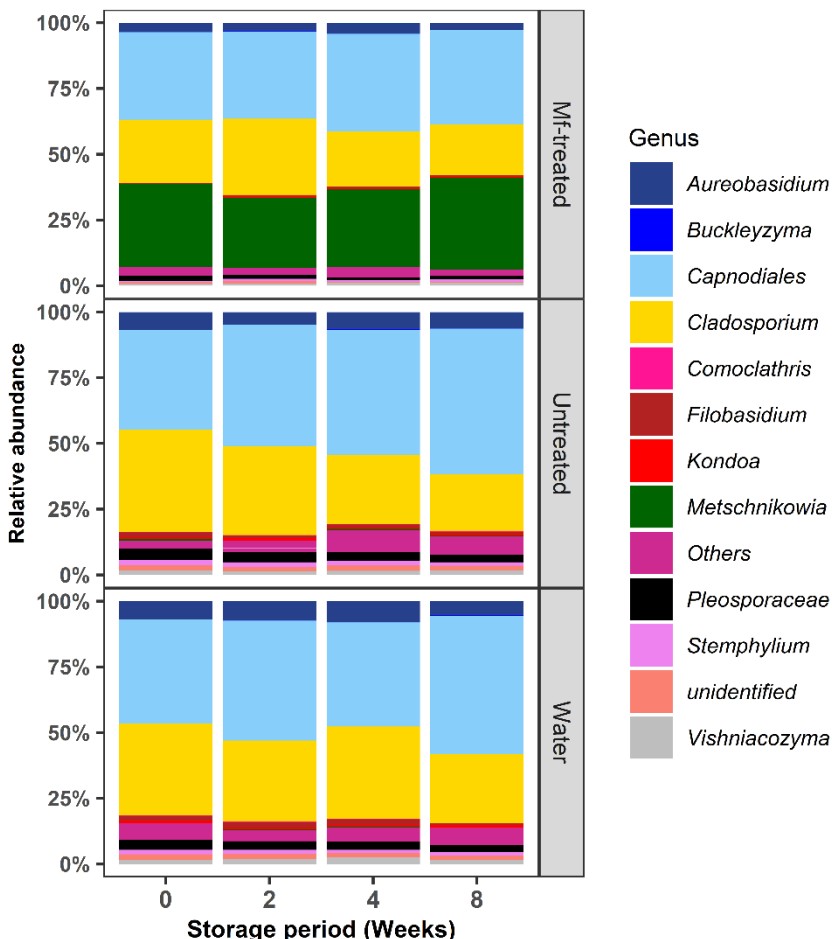

**Figure 3.** Barplots showing the relative abundances of the predominant epiphytic fungal taxa (>0.1%) observed in apples in *Metschnikowia fructicola*-treated and untreated fruit at different storage times. Taxa with relative abundances of less than 0.1% are grouped as 'Others'.

### 3.3. M. fructicola Treatment Reduced the Abundance of Postharvest Fungal Genera during Storage

A specific effort was made to determine the impact of the application of *M. fructicola* on the abundance of fungal postharvest pathogens on apples during storage. Specifically, fungal genera that are known to cause postharvest diseases in apples were identified in our data set and their relative abundance was compared between the *M. fructicola*-treated samples and the untreated and water-treated controls (Figure 4). The results revealed that the relative abundance of ASVs corresponding to the fungal pathogenic genera, including *Alternaria*, *Aspergillus*, *Comoclatris*, *Nigrospora*, *Penicillium*, *Podosphaera,* and *Stemphylium*, was significantly lower in *M. fructicola*-treated samples, relative to water-treated and untreated fruit.

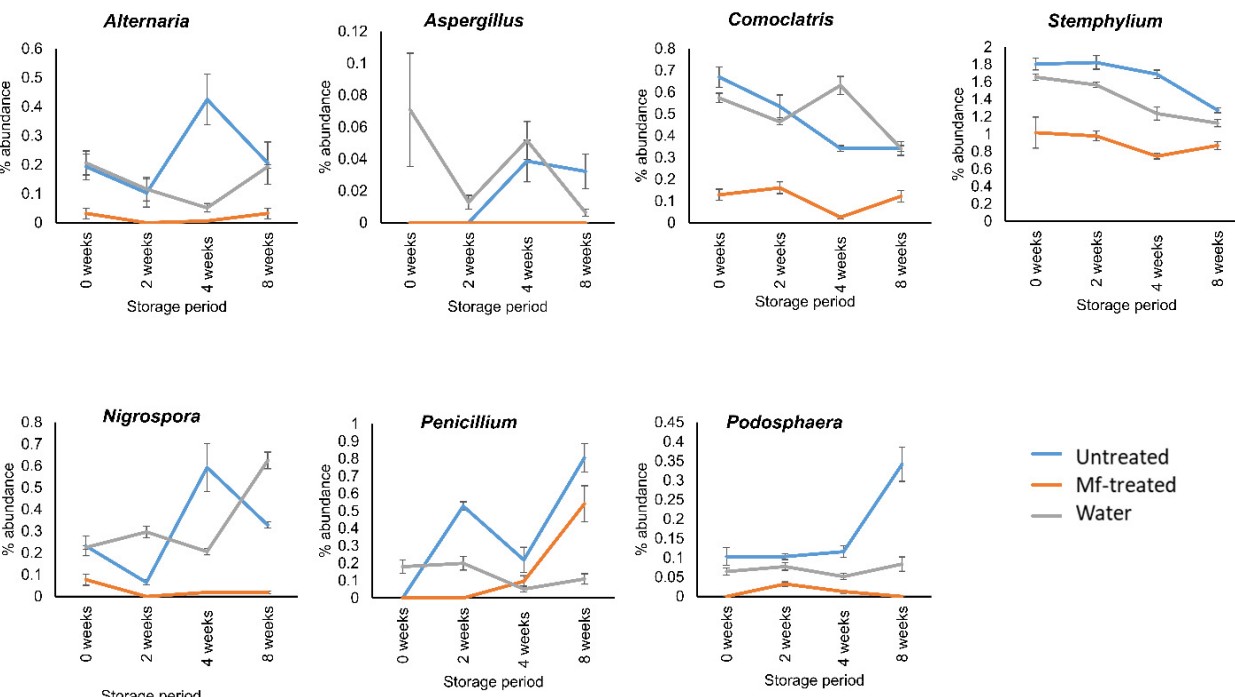

**Figure 4.** Line graphs showing the reduced relative abundances of some postharvest pathogenic genera in *Metschnikowia fructicola* treated fruits as compared with untreated controls.

## 4. Discussion

In the present study, the influence of the application of the yeast biocontrol agent *M. fructicola* on the epiphytic fungal community of apple fruit during storage was evaluated. The application of *M. fructicola* resulted in shifts in fungal community characterized by a significant decrease in diversity (Figure 2). In this regard, Zhimo et al. [5] reported a similar reduction in the richness of the fungal community in strawberry fruit treated with *M. fructicola*. The reduction in fungal diversity following the application of biological control agents has also been reported in other crops, such as tea (*Camellia sinensis*) leaves, as a result of the application of *Serendipita* (=*Piriformospora*) *indica* [24], and in the rhizosphere of chrysanthemum (*Chrysanthemum morifolium*) following the application of *Bacillus subtilis* NCD-2 [25]. In contrast, under natural conditions of cold storage, the diversity of microbes has been shown to increase with time [26], and our result also showed marginal increases in diversity with an increase in storage time in untreated fruit as compared with water-treated or biocontrol-treated fruit, where the diversity remained stable.

The results from our analyses indicated a significant dissimilarity in the overall community composition between *M. fructicola*-treated and control (untreated and water-treated) fruit (Figure 2). Shifts in the apple microbial community due to different management practices have been previously reported, where clear differences in the composition of the mycobiome were observed between organic and conventionally grown apples [27,28]. Postharvest shifts in the apple microbial community have been reported during storage, following the application of waxes, fungicides, and disinfectants, and other physical management practices [28]. The functional implications of these microbial shifts and community alterations as a result of postharvest management practices remain to be elucidated.

Analysis of the most abundant taxa at the genera level identified a total of 12 fungal genera that dominated the fungal community (Figure 3), which was in line with previous reports on the apple microbiome [29]. The high abundance of the applied biocontrol agent *M. fructicola* on the fruit surface was consistent throughout the cold storage period (Figure 3). This persistence implies a prolonged and durable protection of treated fruit against postharvest pathogenic fungi. In this regard, a reduced relative abundance was observed for the genera *Alternaria*, *Aspergillus*, *Comoclatris*, *Nigrospora*, *Podosphaera*, and

*Stemphylium* in yeast-treated fruit relative to the untreated and water-treated controls (Figure 4). As previously noted, several mechanisms of action are exhibited by *M. fructicola* against postharvest pathogens. Our present data indicate that shifts in the fruit mycobiome that result from the application of *M. fructicola* may be considered an additional mechanism for suppressing postharvest pathogens.

Overall, our findings confirm that high-throughput sequencing technologies can be utilized to better understand the interactions between *M. fructicola* with the postharvest epiphytic mycobiome of apple fruit during storage. Our data indicate that the application of *M. fructicola* reduces the abundance of fungal pathogens involved in postharvest decay of apples. The study also revealed that the yeast application causes a shift in the structure and composition of the overall fungal microbiome, potentially resulting in further beneficial effects by creating associations with other fungal species. Further studies to explore the complex microbial interactions that occur during shifts in the structure of microbial communities (bacteria and fungi) are needed to improve the efficacy of biocontrol strategies for the management of postharvest diseases of apple fruit.

**Author Contributions:** Conceptualization, S.D.; methodology, A.B., V.Y.Z., A.K., S.S., and O.F.; software, A.B., V.Y.Z., and A.A.; validation, A.B., V.Y.Z., S.D., and M.W.; formal analysis, A.A., A.B., and V.Y.Z.; investigation, A.B., V.Y.Z., A.K., S.S., and O.F.; resources, S.D.; data curation, A.B., V.Y.Z., and A.A.; writing—original draft preparation, A.B.; writing—review and editing, V.Y.Z., S.D., and M.W.; visualization, A.B. and V.Y.Z.; supervision, S.D.; funding acquisition, S.D. All authors have read and agreed to the published version of the manuscript.

**Funding:** This research was funded by BARD, IS-US Binational Agricultural Research and Development, grant number IS-5455-21, awarded to Samir Droby.

**Data Availability Statement:** The data that support the findings of this study are available on request from the corresponding author.

**Conflicts of Interest:** The authors declare no conflict of interest. The funders had no role in the design of the study; in the collection, analyses, or interpretation of data; in the writing of the manuscript; or in the decision to publish the results.

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
