# Peer review of "Changes in the Fungal Community Assembly of Apple Fruit Following Postharvest Application of the Yeast Biocontrol Agent Metschnikowia fructicola"

_horticulturae, doi:10.3390/horticulturae7100360_

Round 1
Reviewer 1 Report
The manuscript "Changes in the fungal community assembly of apple fruit following postharvest application of the yeast biocontrol agent Metschnikowia fructicola" presents the results of using M. fructicola in biocontrol. The application of biocontrol is a very promising topic, and the presented research broadens the existing knowledge.
Detailed comments:
line 13 - twice is the word Abstract.
The abstract requires some editing and more details.
The article lacks the results of the influence of the strain used for biocontrol on physical and chemical properties, such as e.g. acidity, weight, color. I suggest presenting the results of such research in the next article.
In the figures, the names of microorganisms should be italicized.
The discussion needs to be expanded, adding more previous results and presenting what was new in this research.
If the authors have preserved isolated strains, it is worth checking the effect of M. fructicola on isolates in vitro.
Author Response
Response to reviewer’s comments
The manuscript "Changes in the fungal community assembly of apple fruit following postharvest application of the yeast biocontrol agent Metschnikowia fructicola" presents the results of using M. fructicola in biocontrol. The application of biocontrol is a very promising topic, and the presented research broadens the existing knowledge.
Response: We thank you for your valuable comments and present below our point by point responses to your comments and suggestions.
Comment: line 13 - twice is the word Abstract.
Response: Removed the word Abstract occurring twice in line 13
Comment: The abstract requires some editing and more details.
Response: Edited the abstract including more details of the results.
Comment: The article lacks the results of the influence of the strain used for biocontrol on physical and chemical properties, such as e.g. acidity, weight, color. I suggest presenting the results of such research in the next article.
Response: Thank you for the insight and we will certainly add these results in future articles.
Comment: In the figures, the names of microorganisms should be italicized.
Response: The names of microorganisms has been italicized in Figure 3.
Comment: The discussion needs to be expanded, adding more previous results and presenting what was new in this research.
Response: The discussion has been expanded as suggested.
Comment: If the authors have preserved isolated strains, it is worth checking the effect of M. fructicola on isolates in vitro.
Response: We duly note the suggestion and incorporate these studies in future studies

Reviewer 2 Report
This is a rather short paper presenting some preliminary data on microbial shift on apple surfaces after application of a biocontrol agent. Still, the paper is quite interesting because it adds some much needed information on the efficacy of biocontrol of a very important crop. I recommend publications after some minor edits and improvements.
- The authors don’t present any data on disease symptoms of MF-treated, untreated and water treated apples after 8 weeks storage. Clearly, the focus of the paper lies on microbial community shift, but still it has to be considered that the ultimate goal of biological control is the reduction of diseases, and therefore monitoring of symptom development should always be part of such a study.
- Line 160 ff. According to the authors no effect on storage period was observed within any of the three treatment groups. However, the boxplot for the untreated control (Figure 1 B) shows a rather strong increase of ASV 2, 4 and 8 weeks after treatment. Please explain. The boxplot for water treated apples shows higher levels at 0 weeks after treatment compared to the untreated control. In general, I would expect water treatment to decrease ASV numbers due to the wash off effect. Please discuss.
Some minor edits:
Line 27: Microorganisms represent a fundamental aspect of fruit… Needs clarification! MO live in or on fruits but are not parts of the fruit.
Line 58: Zhimo et al. demonstrated …microbial communities in the fruit. I’m wondering if they analyzed the communities on the fruit (outside) or in (inside) the fruit.
Line 236: M. fruticola should not be underlined.
Author Response
Response to reviewer’s comments
This is a rather short paper presenting some preliminary data on microbial shift on apple surfaces after application of a biocontrol agent. Still, the paper is quite interesting because it adds some much needed information on the efficacy of biocontrol of a very important crop. I recommend publications after some minor edits and improvements.
Response: We thank you immensely for your valuable comments and suggestions. We provide below a point by point responses to the comments from your side.
Comment: The authors don’t present any data on disease symptoms of MF-treated, untreated and water treated apples after 8 weeks storage. Clearly, the focus of the paper lies on microbial community shift, but still it has to be considered that the ultimate goal of biological control is the reduction of diseases, and therefore monitoring of symptom development should always be part of such a study.
Response: As indicated, the focus of the paper was on the microbial dynamics and so we present here only the microbiome results. However, the reviewer is spot on in pointing out the need to monitor disease developments and we shall certainly follow it in future studies.
Comment: Line 160 ff. According to the authors no effect on storage period was observed within any of the three treatment groups. However, the boxplot for the untreated control (Figure 1 B) shows a rather strong increase of ASV 2, 4 and 8 weeks after treatment. Please explain. The boxplot for water treated apples shows higher levels at 0 weeks after treatment compared to the untreated control. In general, I would expect water treatment to decrease ASV numbers due to the wash off effect. Please discuss.
Response: Under undisturbed conditions (untreated), the diversity of microbiomes is be expected to increase with time on the fruit surface during storage as reported in some studies (https://doi.org/10.21203/rs.3.rs-567181/v1, https://doi.org/10.3390/microorganisms8060944) and also seen in our results where we see marginal increases at 2, 4 and a slight decrease at 8 weeks. However, these changes were not statistically significant (p = 0.056). In the case of MF treated and water treated fruit, the diversity (richness) remained more stable than the untreated fruit. We indeed see marginally higher ASVs in washed than unwashed sample, but the difference was not statistically significant either. We postulate that the marginal increase may have been obtained due to the water dipping procedure of the fruits -causing a prolonged wetting of the surface of the fruit, and allowing for more microorganisms to be harvested/ dislodged from the fruit surface when swabbed.
Comment: Line 27: Microorganisms represent a fundamental aspect of fruit… Needs clarification! MO live in or on fruits but are not parts of the fruit.
Response: Corrected as “Microorganisms represent an important component of fruit….”
Comment: Line 58: Zhimo et al. demonstrated …microbial communities in the fruit. I’m wondering if they analyzed the communities on the fruit (outside) or in (inside) the fruit.
Response: Zhimo et al. analyzed the epiphytic communities and is added to the manuscript as “….epiphytic microbial communities….”
Comment: Line 236: M. fruticola should not be underlined.
Comment: Corrected

Reviewer 3 Report
Dear Authors
This is an interesting study. The manuscript overall is good. The content of the paper could be published in Horticulturae. But, the manuscript requires some reviews. In the manuscript, I pointed out some correction as follow:
Line 13: please remove extra “Abstract”
Line 24: please Change “Metschnikowia fructicola” to “Disease management”.
Line 83-84: Change “Metschnikowia fructicola” to “M. fructicola”.
Line 127-128: please correct this sentence “implementing the d amplicon workflow”.
Line 149: please remove extra “obtain”
Line 166-167: please Change “M. fructicola” to “Metschnikowia fructicola”.
Line 188: please Change “M. fructicola” to “Metschnikowia fructicola”.
Line 209: please Change “M. fructicola” to “Metschnikowia fructicola”.
Line 223-224: please Change “M. fructicola” to “Metschnikowia fructicola”.
Line 229: please mentions reference NO. “Zhimo et al. [??]”
Line 231: Remove extra space at the beginning of sentence.
Line 233: Remove extra space between “of” and “the”
Line 233: Change “Piriformospora indica” to “Serendipita (=Piriformospora) indica”.
Line 236: remove underline “M. fructicola”.
Line 236-237: this section is part of material and methods, remove it. “The impact of the application of M. fructicola on the composition of the epiphytic fungal community was also analyzed by PCoA and PERMANOVA.

Author Response
Response to reviewer’s comments
This is an interesting study. The manuscript overall is good. The content of the paper could be published in Horticulturae. But, the manuscript requires some reviews. In the manuscript, I pointed out some correction as follow:
Response: Thank you for your comments and here are the detailed responses to your important comments and suggestions.
Line 13: please remove extra “Abstract”
Response: Corrected as suggested
Line 24: please Change “Metschnikowia fructicola” to “Disease management”.
Response: Corrected as suggested
Line 83-84: Change “Metschnikowia fructicola” to “M. fructicola”.
Response: Corrected as suggested
Line 127-128: please correct this sentence “implementing the d amplicon workflow”.
Response: Corrected as suggested as “….by implementing the default workflow…”
Line 149: please remove extra “obtain”
Response: Corrected as suggested
Line 166-167: please Change “M. fructicola” to “Metschnikowia fructicola”.
Response: Corrected as suggested
Line 188: please Change “M. fructicola” to “Metschnikowia fructicola”.
Response: Corrected as suggested
Line 209: please Change “M. fructicola” to “Metschnikowia fructicola”.
Response: Corrected as suggested
Line 223-224: please Change “M. fructicola” to “Metschnikowia fructicola”.
Response: Corrected as suggested
Line 229: please mentions reference NO. “Zhimo et al. [??]”
Response: Reference number added “Zhimo et al. [55]”
Line 231: Remove extra space at the beginning of sentence.
Response: Corrected as suggested
Line 233: Remove extra space between “of” and “the”
Response: Corrected as suggested
Line 233: Change “Piriformospora indica” to “Serendipita (=Piriformospora) indica”.
Response: Corrected as suggested
Line 236: remove underline “M. fructicola”.
Response: Corrected as suggested
Line 236-237: this section is part of material and methods, remove it. “The impact of the application of M. fructicola on the composition of the epiphytic fungal community was also analyzed by PCoA and PERMANOVA.
Response: Removed as suggested
